# Regulatory standards and guidance for the use of health applications for self-management in Africa: scoping review protocol

Benard Ayaka Bene [1,2] Sunny Ibeneme,[3] Kayode Philip Fadahunsi [1]
Bala Isa Harri,[4] Nkiruka Ukor,[5] Nikolaos Mastellos,[6] Azeem Majeed,[1] Josip Car[1]

## ABSTRACT

**Introduction** Despite health applications becoming ubiquitous and with enormous potential to facilitate self-management, regulatory challenges such as poor application quality, breach of data privacy and limited interoperability have impeded their full adoption. While many countries now have digital health-related policies/strategies, there is also a need for regulatory standards and guidance that address key regulatory challenges associated with the use of health applications. Currently, it is unclear the status of countries in Africa regarding regulatory standards and guidance that address the use of health applications.

This protocol describes the process of conducting a scoping review which aims to investigate the extent to which regulatory standards and guidance address the use of health applications for self-management within the WHO African Region countries.

**Methods** The review will follow the methodological framework for conducting a scoping study by Arksey and O'Malley (2005), and the updated methodological guidance for conducting a Joanna Briggs Institute (JBI) scoping review. Given that regulatory standards and guidance are unlikely to be available in scientific databases, we will search Scopus, Google, OpenGrey, WHO Regional Office for Africa Library (AFROLIB), African Index Medicus (AIM), websites of WHO, ITU and Ministries of Health, repositories for digital health policies. We will also search the reference lists of included documents, and contact key stakeholders in the region. Results will be reported using descriptive qualitative content analysis based on the review objectives. The policy analysis framework by Walt and Gilson (1994) will be used to organise findings. A summary of the key findings will be presented using tables, charts and maps.

**Ethics and dissemination** The collection of primary data is not anticipated in this study and hence ethical approval will not be required. The review will be published in a peer-reviewed journal while key findings will be shared with relevant organisations and/or presented at conferences.

For numbered affiliations see end of article.

**Correspondence to**
Dr Josip Car;
josip.car@imperial.ac.uk

## Strengths and limitations of this study

► This study, to the best of our knowledge, is the first attempt to identify gaps in the health application regulatory environment across WHO African Region countries.
► The study will also map out the key stakeholders and their roles in regulating the use of health applications for patient self-management across WHO African Region countries.
► Considering that regulatory standards and guidance may not be publicly available, it is possible that some documents may be missed; however, efforts will be made to obtain these documents by contacting key stakeholders.

their healthcare providers to maintain good health.[1 2] As the world experiences a massive proliferation of mobile applications, better internet access and advances in artificial intelligence, it is increasingly becoming easier for people to engage in self-management.[3–6]

A mobile application, which is a software application designed to run on mobile devices such as smartphones and tablet computers, can provide an important platform for self-management.[3 7] Health applications have the unique ability to facilitate one or more aspects of self-management of diseases by capturing the user's health data, and providing tailored information, instructions, graphic displays, guidance and reminders.[8–12] They enable remote monitoring of patient's health status while also linking patients to their healthcare providers and social networks.[8–12] Rather than manually recording the user's health data, some health applications are designed to allow patient's health data such as blood glucose, blood pressure, heart rate and weight to be automatically transmitted to the application (via Bluetooth) from measuring devices, wearables or sensors.[13]

## BACKGROUND
Self-management is a crucial component of chronic disease management, and it incorporates the daily practices carried out by individuals intentionally and with the guidance of

BMJ

The use of health applications for self-management could increase access to and reach (coverage) of effective and safe healthcare services at scale while also promoting autonomy, person-centred care, equity and privacy and confidentiality.[3 14–17] This also presents an opportunity to strengthen health systems in Africa and other low-income and middle-income countries where the potential of these applications has hardly been harnessed. For over a decade, mobile health-related interventions in Africa have been restricted to services that do not require smartphones or internet connection, such as short messaging service (SMS) and use of voice notes.[3 18 19] This is partly because of the limited access to smartphones and low penetration of internet services on the continent. However, this narrative is changing as there is increased availability of cheaper smartphones, and the cost of mobile internet subscription is rapidly declining,[20] thus presenting an immense opportunity to close the digital heath divide that had existed between Africa and high-income countries for decades. The adoption of smartphones in sub-Saharan Africa continues to rise rapidly moving from 10% of total smartphone connections in 2014 to about 50% in 2020. This is expected to rise to 65% by the end of 2025 with about 678 million smartphone connections.[21 22]

Because health applications for self-management present many opportunities and potential for the continent of Africa, appropriate regulatory standards and guidance are required to facilitate the adoption of evidence-based, safe and efficient health applications.[13 23] The WHO affirms that the existence of regulatory standards and guidance play an important role in strengthening digital health governance at both national and international levels.[24 25] They provide a framework and guidance (with clearly defined roles) for data governance that ensures data protection, confidentiality and integrity of personal health data and system accessibility.[25] Globally, the delay in establishing appropriate regulatory standards and guidance has led to the persistence of major regulatory challenges including poor application quality,[26–30] breach of data privacy[31 32] and limited interoperability,[13 33–35] thus impeding the maturity of health applications (as a fully integrated component of the healthcare system).[13] Africa, however, has a golden opportunity of addressing these challenges early and thus leapfrogging the barriers that many high-income countries have had to deal with.[36]

Many countries including those in Africa now have some form of national policies or strategies on digital health.[25] However, beyond having national policies and strategies on digital health, it is imperative for countries in Africa to have regulatory standards and guidance that address key regulatory challenges affecting the use of health applications. So far, it is unclear which countries in the WHO African Region have such regulatory standards and guidance. The Global Digital Health Index and Maturity Model, an initiative of HealthEnabled and its partners, was launched in 2018 to serve as an interactive digital platform for tracking, monitoring and evaluating the use of

digital health technologies across different countries.[37 38] The inaugural report on the Global Digital Health Index was subsequently published in 2019.[37] To the best of our knowledge, this inaugural report is the first attempt to evaluate countries' progress in digital health using 19 core indicators grouped under seven categories of key indicators of the WHO/ITU's National eHealth Strategy Toolkit (leadership and governance; strategy and investment; legislation, policy and compliance; workforce; standards and interoperability; infrastructure; and services and applications). Although the report provided valuable insights into the current state of legislation, policy and compliance as well as standards and interoperability across countries, only 6 of 47 countries (Benin, Ethiopia, Mali, Nigeria, Sierra Leone, and Uganda) in the WHO African Region participated in the evaluation. Moreover, the report did not provide any insight on whether countries have standalone regulatory standards and guidance on digital health and whether existing policies and strategies address key regulatory challenges related to the use of health applications for self-management.

In this paper, we present a protocol for a scoping review to investigate the extent to which regulatory standards and guidance address key regulatory challenges affecting the use of health applications for self-management in countries (Algeria, Angola, Benin, Botswana, Burkina Faso, Burundi, Cameroon, Cape Verde, Central African Republic, Chad, Comoros, Ivory Coast, Democratic Republic of the Congo, Equatorial Guinea, Eritrea, Ethiopia, Gabon, Gambia, Ghana, Guinea, Guinea-Bissau, Kenya, Lesotho, Liberia, Madagascar, Malawi, Mali, Mauritania, Mauritius, Mozambique, Namibia, Niger, Nigeria, Republic of the Congo, Rwanda, São Tomé and Príncipe, Senegal, Seychelles, Sierra Leone, South Africa, South Sudan, Eswatini, Togo, Uganda, Tanzania, Zambia and Zimbabwe) within the WHO African Region.

## AIM AND OBJECTIVES
### Aim
To identify and assess the extent to which regulatory standards and guidance regulate the use of health applications for self-management, and to map out the key stakeholders and their responsibilities in regulating the use of health applications for patient self-management across WHO African Region countries.

### Objectives
The objectives of this review are to:
1. *Identify* regulatory standards and guidance that are available for regulating the use of health applications for self-management across WHO African Region countries.
2. *Assess* the extent to which the regulatory standards and guidance regulate the use of health applications for self-management in terms of what aspects are regulated, why, how and for whom, and what aspects are not regulated.

3. *Map* out the key stakeholders and their roles in regulating the use of health applications for patient self-management.

## METHODS
### Study design
A team comprising experts and researchers with experience and interest in the relevant disciplines (digital health, self-management, health policy and planning and scoping review methodology) will design, conduct and report the scoping review. The process of the scoping review will follow the methodological framework for conducting a scoping study originally described by Arksey and O'Malley (2005), and the updated methodological guidance for conducting a Joanna Briggs Institute (JBI) scoping review.[39–42] While stage 1 of the framework will guide the formation of the review question and search strategy, stage 2 and stage 3 will subsequently guide the identification of relevant studies and the study selection process respectively. Stage 4 will guide the development of the data-charting form as well as the data charting (extraction) process itself, whereas stage 5 will guide the process of collation, summarisation and reporting of results. The final stage (stage 6) is optional, and it is meant to guide the process of consultation with stakeholders (if required). The reporting of the scoping review will be guided by the Preferred Reporting Items for Systematic Reviews and Meta-Analyses extension for Scoping Reviews (PRISMA-ScR) checklist.[43] The PRISMA-ScR checklist will be completed and attached as an online supplemental additional file.

This review is not registered with PROSPERO or any similar registration platform as it is not a requirement for scoping reviews to be preregistered.[42]

### Stage 1: identifying the research questions
#### Research questions
The scoping review will attempt to answer the following research questions, which were generated based on gaps identified from preliminary literature review:
1. What regulatory standards and guidance are available for regulating the use of health applications for self-management across WHO African Region countries?
2. To what extent do regulatory standards and guidance regulate the use of health applications for self-management in terms of what aspects are regulated, why, how and for whom, and what aspects are not regulated?
3. Who are the key stakeholders and what are their roles in regulating the use of health applications for patient self-management?

### Stage 2: identifying relevant documents
A comprehensive search strategy will be designed by two reviewers with the assistance of a librarian and in consultation with other research team members. The following key terms will be included: policy, legislation, strategy, regulation, standard, criterion, framework, guidance, guideline, digital health, eHealth, app, WHO African Region and sub-Saharan Africa, and list of all countries within the WHO African Region. We will use truncation to increase yield of results.

The search strategy will then be used to search for all eligible documents and policy related sources of evidence including conference proceedings and grey literature (committee reports and government reports). Boolean terms (mainly 'OR' and 'AND') will be used to combine search results. 'OR' will be used to combine results within the same category while 'AND' will be used to combine result between different categories to narrow down the search results.

Considering that regulatory standards and guidance are unlikely to be available in scientific databases, we will focus our search on Scopus, Google, OpenGrey, WHO Regional Office for Africa Library (AFROLIB), African Index Medicus (AIM), websites of WHO, ITU and Ministries of Health, and repositories for digital health policies (including ICT Works, WHO directory of eHealth policies and HIS Strengthening Resource Centre). In addition, we will search the reference lists of included documents, and contact key stakeholders in the region, including persons working at the Ministry of Health (or other relevant departments and institutions), WHO Country Offices and the WHO Regional Office for Africa.

A sample search strategy for Scopus is provided in online supplemental file 1.

### Stage 3: study selection
Standalone regulatory standards and guidance; national policies and strategies; and other sources of evidence including committee or government reports (published or unpublished) identified from searches and key contact persons will be imported into Mendeley reference management software to remove duplicates. After removing duplicates, the remaining documents will be imported into Covidence (a web-based tool that allows two reviewers to independently manage article screening and data extraction). Using the predefined eligibility criteria outlined below, two reviewers, working independently, will screen each source of evidence in two stages (title and abstract, and full text) for possible inclusion in the review. Any discrepancies will be discussed in order to reach a consensus. Unresolved disagreements will be further deliberated on and resolved in a steering group meeting involving a third reviewer.

### Criteria for selection of documents for this review
#### Inclusion criteria
The following categories of documents that will be considered for inclusion in this review:
► Standalone regulatory standards and guidance that potentially regulate or provide guidance for the use of health applications.
► National policies and strategies on digital health developed and produced by countries within the WHO African Region.

► Other sources of evidence (published or unpublished) including committee or government reports that potentially address regulatory issues related to the use of health applications in any WHO African Region country.

### Exclusion criteria

Documents that fall within the following categories will be excluded:

► Regulatory standards, guidance, policies, strategies and committee or government reports that address regulatory issues related to the use of health applications for patient self-management in countries outside the WHO African Region.

► Documents that were written or produced earlier than 2005. This is because it was only in the year 2005 that global efforts toward promoting standards to minimise variability and potential harms that could arise from poorly regulated use of digital health began.[25]

### Stage 4: charting (extraction) the data

Two reviewers in consultation with the other members of the research team will develop the data extraction form using an iterative process. At the initial stage of this process, a pilot data extraction will be carried out on at least 10% of included sources of evidence. The form will continuously be refined until a consensus is reached.

Following the development of the data extraction form, full data extraction will then be done by the two reviewers independently and any disagreement will be resolved by discussion. Inconsistencies or unresolved issues will be discussed and resolved with a third reviewer in a steering group meeting. Any missing information that is relevant to this review will be sought from the relevant stakeholders.

As recommended by Peters *et al*,[42] the data to be extracted will include author (s); year of production or publication; country (where the document was produced or published); type of document (eg, standalone regulatory standard/guidance, national policy, other sources of evidence including government reports); document development approach (eg, public–private/multisectoral approach or private sector approach); the aspects of health applications for self-management that are regulated; the reasons why they are regulated; how the regulations are enforced; for which population are the regulations targeted (eg, applications users, application developers and healthcare providers); and the aspects of health applications for self-management that are not regulated; and the key stakeholders and their responsibilities in regulating the use of health applications for patient self-management.

### Stage 5: collating, summarising and reporting the results

Unlike systematic reviews, the goal of scoping reviews is not to synthesise the results or outcomes of included sources of evidence. However, as recommended by Peters *et al*, the scoping review authors may choose to extract results and descriptively (rather than analytically) map them.[42]

Therefore, data extracted in stage 4 will be reported using descriptive qualitative content analysis based on the review objectives and as an attempt to answer the research questions. The policy analysis framework by Walt and Gilson[44] will be used to organise findings according to the aspects of the use of health applications for self-management that are regulated, the reasons why they are regulated, how the regulations are enforced, for which population are the regulations targeted. This will be done based on four priori categories, namely: content, context, process and actors. The Reporting Items for Stakeholder Analysis (ie, the RISA tool)[45] will be used to guide the mapping of key stakeholders and their responsibilities in regulating the use of health applications for patient self-management.

Following descriptive qualitative content analysis, summary of the key findings will be presented using tables, charts and geospatial map(s) to aid visualisation.[46] More specifically, geospatial mapping will help highlight gaps in coverage of regulatory standards and guidance for regulating the use of health applications for self-management across countries within the WHO African Region.

### Stage 6: consultation

Stage 6 (consultation) is an optional but important component of scoping study methodology.[41] While this study does not include stage 6, there are future plans for consultation with patients, healthcare professionals and policymakers in form of interviews, focus group discussions and stakeholders' workshop.

## PATIENT AND PUBLIC INVOLVEMENT

There was no direct involvement of patients and the public in the design of this study. Nevertheless, the interests of patients and the public were among the major factors that informed the study design.

## ETHICS AND DISSEMINATION

This study is mainly a review with no intention to collect primary data. Hence, application for ethical approval is not required.

On completion of the study, a manuscript will be submitted to a peer-reviewed journal for publication while summary of the findings will be shared with relevant organisations. Key findings will be summarised and presented at national, regional and international conferences.

**Author affiliations**
[1]Department of Primary Care and Public Health, Imperial College London, London, UK
[2]Department of Public Health, Federal Ministry of Health, Abuja, Federal Capital Territory, Nigeria
[3]Office of the Assistant Regional Director, Research, Development & Innovations Division, WHO – African Regional Office, Brazzaville, Congo
[4]Department of Health Planning, Research and Statistics, Federal Ministry of Health, Abuja, Federal Capital Territory, Nigeria

[5]Universal Health Coverage/Live course Cluster (UHC/LC), WHO – Nigeria Country Office, Abuja, Federal Capital Territory, Nigeria
[6]School of Public Health, Imperial College London, London, UK

Acknowledgements We appreciate Rebecca Jones, the Library Manager and Liaison Librarian at the Charing Cross Library, Imperial College London, for her advice and assistance with the search strategy for the main review.

Contributors BAB and JC conceived the study. BAB designed the study with contributions from JC and NM. BAB drafted the manuscript and JC, NM, AM, SI, KPF, BIH and NU read and contributed to it. AM is the clinical lead while JC acts as a guarantor for the study. The final manuscript was read and approved by all the authors.

Funding This work is part of BAB's PhD research which is sponsored by the Nigerian government. AM and JC are supported by NIHR Applied Research Collaboration North-West London, grant number NIHR200180. The views expressed in this publication are those of the authors and not necessarily those of the Nigerian government or the NIHR or the Department of Health and Social Care.

Competing interests None declared.

Patient consent for publication Not applicable.

Provenance and peer review Not commissioned; externally peer reviewed.

ORCID iDs
Benard Ayaka Bene http://orcid.org/0000-0003-4545-1836
Kayode Philip Fadahunsi http://orcid.org/0000-0003-1470-5493

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
