## [Reviewer comments · BMJ Open]

ARTICLE DETAILS

TITLE (PROVISIONAL)	Regulatory Standards and Guidance for the Use of Health Apps for Self-Management in Africa: Scoping Review Protocol
AUTHORS	Bene, Benard; Ibeneme, Sunny; Fadahunsi, Kayode; Harri, Bala; Ukor, Nkiruka; Mastellos, Nikolaos; Majeed, Azeem; Car, Josip

VERSION 1 – REVIEW

REVIEWER	CShyam Institute haurasia, Alok Ranjan
REVIEW RETURNED	04-Nov-2021

GENERAL COMMENTS	The paper identifies three categories of regulatory systems: 1) standalone regulatory systems; 2) National policies and strategies; and 3) written or published evidence. It is important that the selection criteria takes into consideration the three categories in the sense that the scoping review must have appropriate representation of the three categories of regulatory systems.
--

REVIEWER	Diggle, Peter Lancaster University, Faculty of Health and Medicine
REVIEW RETURNED	09-Nov-2021

GENERAL COMMENTS	A generally very clear account of the protocol for a largely (and appropriately so) qualitative investigation into an important problem - the proliferation of health apps of dubious quality and the consequent need for regulatory standards. Also, a clear rationale for basing the study in WHO's African region. Minor pints of clarification are listed in th attachment to this report.
--

REVIEWER	Nittas, Vasileios University of Zurich, Epidemiology, Biostatistics and Prevention Institute
REVIEW RETURNED	17-Nov-2021

GENERAL COMMENTS	This protocol describes a scoping review the aims to identify and compare standards and guidance for regulating the use of health apps for self-management within the WHO African Region. While the research question is timely and important, I have a few major concerns regarding whether a scoping review approach (as currently described) is the most practical/feasible to answer it. Please find my thoughts below Major remarks:
--

	Page 14: “Stage 1: Identifying the research question” – this section is rather meant for describing how the research question was formed. E.g. through expert consultations, preliminary literature screenings and the identification of gaps etc.,. Please add 1-2 sentences of how the questions came to be. For the same section as above: the inclusion and exclusion criteria rather below in the study section (stage 2) and not under stage 1. Page 15: Stage 2: I find the description of the search strategy somehow unclear and potentially impractical, for several reasons a) In the inclusion criteria you mention that the target of the review are policy documents. If that is true, then scientific databases (such as Medline etc.) are not the right source, as these mostly publish peer reviewed scientific literature b) If you decide to also include scientific papers, then you might need to change the review’s entire focus, as regulatory standards are rarely reported in scientific publications c) I do think that policy documents (grey literature) is the way to go for answering your questions, however, that requires a more iterative process of online/manual searches and is unlikely to be achieved through traditional database searches, as you currently describe. Minor remarks: I personally think that the introduction is too long. I would suggest shortening the "evolution of digital health: intro part, as all that is well-known and extensively published about. As a reader, I can suggest limiting the introduction to those aspects that are most relevant for the addressed topic, keeping it short and to-the-point.
--	---

VERSION 1 – AUTHOR RESPONSE

Reviewer 1

Comment: The paper identifies three categories of regulatory systems: 1) standalone regulatory systems; 2) National policies and strategies; and 3) written or published evidence. It is important that the selection criteria take into consideration the three categories in the sense that the scoping review must have appropriate representation of the three categories of regulatory systems.

Response: The study selection will take into account the three categories of documents. The statement “Standalone regulatory standards and guidance; national policies and strategies; and other sources of evidence including committee or government reports (published or unpublished)” is now added to the study selection section to indicate that the three categories of documents are being considered.

Reviewer 2

Comment 1: Page 4, the statement “hence ethical approval will not be required” is partially contradicted later in the paper (page 18 “Application for ethical approval will be made to the Imperial College Research Ethics Committee...if required”

Response: Stage 6: Consultation section has been revised to read “Stage 6 (consultation) is an optional but important component of scoping study methodology (41). While this study does not include stage 6, there are future plans for consultation with patients, healthcare professionals and policy makers in form of interviews, focus group discussions and stakeholders’ workshop.”

Comment 2: Pages 6 to 12, the background section seems rather long for a study protocol paper, but I have no criticisms of it on its own terms – for editorial decision

Response: The introduction/background has been revised and shortened.

Comment 3: Page 17, “Summary of the key findings will be presented using tables, charts, and geospatial map(s).” I would have welcomed a little more detail here. It’s not clear to me whether any statistical analysis is intended, or what exactly will be mapped. I’m guessing (but shouldn’t have to) that the mapping is intended to describe the Africa-wide coverage of something, but I don’t know what the something is, or at what spatial resolution (country, district, sub-district,...) , or maybe it’s to highlight gaps in coverage, or something else?

Comment 4: This statement has been added “Following descriptive qualitative content analysis, summary of the key findings will be presented using tables, charts, and geospatial map(s) to aid visualisation (46). More specifically, geospatial mapping will help highlight gaps in coverage of regulatory standards and guidance for regulating the use of health apps for self-management across countries within the WHO African Region.”

Comment 5: Page 18, “there is a plan to consult with stakeholders (including patients) if additional information is required.” Yes, but what is the substance of the plan, and who decides if it is required? I would expect at least an outline plan for PPI to be formulated from the outset.

Response: The statement has been revised to read “There was no direct involvement of patients and the public in the design of this study. Nevertheless, the interests of patients and the public were among the major factors that informed the study design.”

Reviewer 3

Comment 1: “Stage 1: Identifying the research question” – this section is rather meant for describing how the research question was formed. E.g. through expert consultations, preliminary literature screenings and the identification of gaps etc. Please add 1-2 sentences of how the questions came to be.

Response: This statement has been revised to read “The scoping review will attempt to answer the following research questions, which were generated based on gaps identified from preliminary literature review.”

Comment 2: For the same section as above: the inclusion and exclusion criteria rather below in the study section (stage 2) and not under stage 1.

Response: The inclusion and exclusion criteria have now been moved to the Study Selection section.

Comment 3: Page 15: Stage 2: I find the description of the search strategy somehow unclear and potentially impractical, for several reasons

a) In the inclusion criteria you mention that the target of the review are policy documents. If that is true, then scientific databases (such as Medline etc.) are not the right source, as these mostly publish peer reviewed scientific literature

b) If you decide to also include scientific papers, then you might need to change the review’s entire focus, as regulatory standards are rarely reported in scientific publications

c) I do think that policy documents (grey literature) is the way to go for answering your questions, however, that requires a more iterative process of online/manual searches and is unlikely to be achieved through traditional database searches, as you currently describe.

Response: This section has been revised to read “Considering that regulatory standards and guidance are unlikely to be available in scientific databases, we will focus our search on Scopus, Google, OPENGREY, WHO Regional Office for Africa Library (AFROLIB), African Index Medicus (AIM), websites of WHO, ITU and Ministries of Health, and repositories for digital health policies (including ICT Works, WHO directory of eHealth policies and HIS Strengthening Resource Centre). In addition, we will search the reference lists of included documents, and contact key stakeholders in the region, including persons working at the Ministry of Health (or other relevant departments and institutions), WHO Country Offices, and the WHO Regional Office for Africa.”

Comment 4: I personally think that the introduction is too long. I would suggest shortening the “evolution of digital health: intro part, as all that is well-known and extensively published about. As a

reader, I can suggest limiting the introduction to those aspects that are most relevant for the addressed topic, keeping it short and to-the-point.

Response: The introduction has been revised and shortened